# Personalized Prediction of Future Lesion Activity and Treatment Effect in Multiple Sclerosis from Baseline MRI

Joshua Durso-Finley[1]                                    JDDURSOF@CIM.MCGILL.CA
Jean-Pierre R. Falet[1,2]                                   JPFALET@CIM.MCGILL.CA
Brennan Nichyporuk[1]                                   BRENNANN@CIM.MCGILL.CA
Douglas L. Arnold[2]                               DOUGLAS.ARNOLD@MCGILL.CA
Tal Arbel[1]                                                   ARBEL@CIM.MCGILL.CA

[1] *Centre for Intelligent Machines, Department of Electrical and Computer Engineering, McGill University & MILA Quebec AI Institute, Canada*

[2] *Montreal Neurological Institute, McGill University, Montreal, Canada*

**Editors:** Under Review for MIDL 2022

## Abstract

Precision medicine for chronic diseases such as multiple sclerosis (MS) involves choosing a treatment which best balances efficacy and side effects/preferences for individual patients. Making this choice as early as possible is important, as delays in finding an effective therapy can lead to irreversible disability accrual. To this end, we present the first deep neural network model for individualized treatment decisions from baseline magnetic resonance imaging (MRI) (with clinical information if available) for MS patients. Our model (a) predicts future new and enlarging T2 weighted (NE-T2) lesion counts on follow-up MRI on multiple treatments and (b) estimates the conditional average treatment effect (CATE), as defined by the predicted future suppression of NE-T2 lesions, between different treatment options relative to placebo. Our model is validated on a proprietary federated dataset of 1817 multi-sequence MRIs acquired from MS patients during four multi-centre randomized clinical trials. Our framework achieves high average precision in the binarized regression of future NE-T2 lesions on five different treatments, identifies heterogeneous treatment effects, and provides a personalized treatment recommendation that accounts for treatment-associated risk (e.g. side effects, patient preference, administration difficulties).

**Keywords:** treatment effect, causal inference, CATE, neuroimaging, precision medicine, multiple sclerosis, new and enlarging lesions, MRI, predicting future outcomes

## 1. Introduction

Precision medicine involves choosing a treatment that best balances efficacy against side effects/personal preference for the individual. In many clinical contexts, delays in finding an effective treatment can lead to significant morbidity and irreversible disability accrual. Such is the case for multiple sclerosis, a chronic neurological disease of the central nervous system. Although numerous treatments are available, each has a different efficacy and risk profile, complicating the task of choosing the optimal treatment for a particular patient. One hallmark of MS is the appearance of lesions visible on T2-weighted MRI sequences of the brain and spinal cord (Rudick et al., 2006). The appearance of new or enlarging, NE-T2, lesions on sequential MRI indicates new disease activity. Suppression of NE-T2 lesions constitutes a surrogate outcome used to measure treatment efficacy. Predicting the

future effect of a treatments on NE-T2 lesions counts using brain MRI prior to treatment initiation would therefore have the potential to be an early and non-invasive mechanism to significantly improve patient outcomes.

Predicting future treatment effects first requires accurate prognostic models for future disease evolution. Deep learning has been used to predict prognostic outcomes in a variety of medical imaging domains (González et al., 2018; Nielsen et al., 2018; Lin et al., 2018; Sun et al., 2019). In the context of MS, research has mainly focused on the related tasks of lesion segmentation (Valverde et al., 2017; Roy et al., 2018; Nair et al., 2020; Nichyporuk et al., 2021) and NE-T2 lesion detection (Doyle et al., 2018; Sepahvand et al., 2020). Recently, deep learning models have been developed for the binary prediction of future disability progression (Tousignant et al., 2019) and the binary prediction of future lesion activity (Sepahvand et al., 2019), as defined by the presence of more than one NE-T2 or Gadolinium enhancing lesions. The prediction of more granular outcomes, such as future NE-T2 lesion counts, remains an open research topic. Furthermore, models are typically built as prognostic models for untreated patients. Predicting prognosis on treatment requires addressing the additional challenge of learning the effect each treatment will have on a particular patient based on their MRI, and thus potentially subtle MRI markers predictive of future treatment response. Machine learning models that have been devised to predict treatment response when it is directly measurable on the image (e.g. shrinking tumour) (Xu et al., 2019; Ha et al., 2018), are insufficient for the context of MS and for other diseases where treatment response must be evaluated relative to placebo or other treatments. Previous work by (Doyle et al., 2017) examined the ability of classical machine learning models to perform binary activity prediction for patients on MS treatments and identify potential treatment responders.

Several machine learning methods have been developed to estimate treatment effects for single treatment-control comparisons (Louizos et al., 2017; Shalit et al., 2017; Shi et al., 2019), with extensions to multiple treatments (Zhao et al., 2017; Zhao and Harinen, 2020). Zhao and Harinen (2020) also integrate the notion of *value* and *cost* (or risk) associated with a treatment, crucial elements for making sound recommendations, particularly when higher efficacy medications may be associated with more severe side effects. However, applications to precision medicine have largely focused on using clinical data as input (Katzman et al., 2018; Fotso, 2018; Ching et al., 2018; Jaroszewicz, 2014). Existing MS models (Sormani et al., 2013; Río et al., 2008; Prosperini et al., 2009) are also limited to clinical features (e.g. demographics), and established group-level MRI-*derived* features (e.g. contrast-enhancing lesion counts, brain volume). Deep learning models would permit learning individual, data-driven features of treatment effect directly from MRI sequences and should provide improvement on existing strategies.

This paper introduces the first image-based treatment recommendation framework for MS that combines prognosis prediction, treatment effect estimation, and treatment-associated risk (Figure 1) evaluation. Our models takes multi-sequence MRI at baseline, along with available clinical information, as input to a multi-head deep neural network that learns shared latent features in a common ResNet encoder (He et al., 2015). It then learns treatment-specific latent features in each output head for predicting future potential outcomes on multiple treatments. Predictions, effect estimates, and treatment risk are then supplied to a *Clinical Decision Support Tool* that outputs a treatment recommendation.

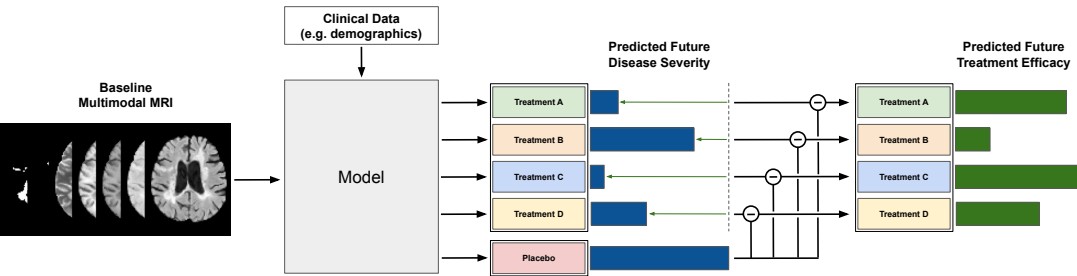

Figure 1: System overview illustrating the overall approach.

This framework is evaluated on a proprietary multi-trial, multi-scanner dataset of MS patients exposed to five different treatment options. The multi-head model not only accurately predicts, from baseline, future NE-T2 lesion counts that will develop 1-2 years ahead on all treatments, but it is able to reliably identify subgroups with heterogeneous treatment effects (groups for which the treatment is more or less effective) as measured by causal inference metrics. Finally, this framework shows that improved lesion suppression can be achieved using the support tool, especially when treatment risk is being considered.

## 2. Method

### 2.1. Estimating Treatment Effect

Let $X \in \mathbb{R}^d$ be the input features (multi-sequence MRI and available clinical data), $Y \in \mathbb{R}$ be the outcome of interest, and $W \in \{0, 1, ..., m\}$ be the treatment allocation in the case where $w = 0$ is a control (e.g. placebo) and the remaining are $m$ treatment options. Given an observational dataset $\mathcal{D} = \{(x_i, y_i, w_i)\}_{i=1}^n$, the individual treatment effect (ITE) for patient $i$ can be defined using the Neyman/Rubin Potential Outcome Framework (Rubin, 1974) as $ITE_i = Y_i(t) - Y_i(0)$, where $Y_i(t)$ and $Y_i(0)$ represents *potential* outcomes on treatment $t \in \{1, ..., m\}$ and control, respectively. The ITE is therefore a fundamentally unobservable causal quantity because only one of these potential outcomes is realized. Treatment effect estimation in machine learning therefore relies on a related causal estimand, the conditional average treatment effect (CATE)

$$\tau_t(x) = \mathbb{E}[Y(t)|X = x] - \mathbb{E}[Y(0)|X = x]. \tag{1}$$

The causal expectations can be recovered from the observational data as follows

$$\tau_t(x) = \mathbb{E}[Y|X = x, W = t] - \mathbb{E}[Y|X = x, W = 0] = \mu_t(x) - \mu_0(x) \tag{2}$$

which can be estimated in an unbiased fashion using randomized control trial data (as in our case), where $\{(Y(0), Y(1))\} \perp\!\!\!\perp W|X$ (Gutierrez and Gérardy, 2017). Further assumptions are needed in the context of non-randomized data (Guelman, 2015).

## 2.2. Network Architecture

Our network is based on TARNET (Shalit et al., 2017) and its multi-treatment extension (Zhao and Harinen, 2020). Specifically, we employ a single multi-head neural network composed of $m$ different CATE estimators,

$$\hat{\tau}_t(x) = \hat{\mu}_t(x) - \hat{\mu}_0(x), \ t \in \{1, ..., m\} \tag{3}$$

where each $\hat{\mu}_t(x)$ is parametrized by a neural network trained on the corresponding treatment distribution, and all share parameters in the earlier layers. A ResNet encoder is used as the shared trunk, and after a global max pooling layer, the encoded features are concatenated with any available clinical information before being processed by treatment-specific multilayer perceptrons (MLPs). The model architecture is depicted in Figure 2.

During training, mini-batches are randomly sampled from $\mathcal{D}$ and fed through the network, outputting a prediction for each treatment head. Losses are computed at each head $t$ for the set of prediction-target pairs where ground truth is available for that treatment, $\{(\hat{y}_{i,t}, y_i)\}_{i:w_i=t}$. Shared parameters are learned in the common layers, which receive gradients for each sample irrespective of treatment allocation, while treatment-specific parameters are learned in the treatment heads from samples allocated to the corresponding treatment. At inference, predictions from all output heads are used for every patient. Full implementation details can be seen in Appendix A.

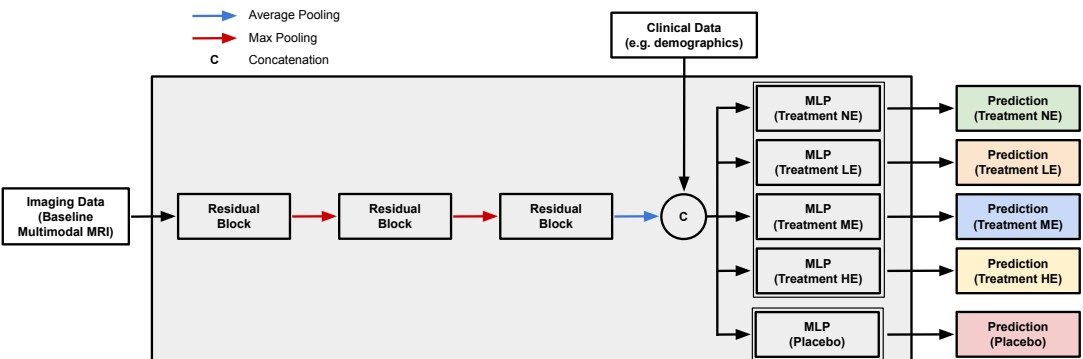

Figure 2: Network Diagram. Common ResNet encoder followed by treatment-specific output MLPs for predicting potential outcomes on multiple treatments.

The tasks of regression and classification are examined. Regressing future NE-T2 lesion counts offers the most intuitive interpretation of treatment effect $\hat{\tau}_t(x)$ (i.e. differences in lesion count), but is sensitive to outliers in the count distribution (e.g. patients with 50 lesions). On the other hand, MS guidelines (Freedman et al., 2020) report a cutoff of ($\geq 3$) new/enlarging T2 lesions after which a treatment should be changed to a more effective one. We therefore also consider the binary classification task of predicting minimal evidence of disease activity on future T2 sequences, referred to as *MEDA-T2*, as having $< 3$ NE-T2 lesions. Unfortunately, the treatment effect $\hat{\tau}_t(x)$ at the binary scale would not capture the true range of effects, and using the softmax outputs to compute $\hat{\tau}_t(x)$ has a less

informative interpretation as compared to regressed counts. For the regression loss, we use Mean Squared Error (MSE) on the log-transformed count, $ln(y_i + 1)$, to reduce the weight of outliers. For the classification loss, we use binary cross entropy (BCE) on the binary MEDA-T2 outcome, $I(y_i < 3)$, where $I(\cdot)$ is the indicator function.

## 2.3. Clinical Decision Support Tool

Based on Zhao and Harinen (2020), we define $r_t$ to be the risk associated with treatment $t \in \{1, 2, ..., m\}$. This can be set by a clinician and patient based on their experience/preference, or could be extrapolated from long-term drug safety data. In the case of MS, drugs can be grouped into lower efficacy (LE), moderate efficacy (ME), and high efficacy (HE). An escalation strategy (starting with LE and escalating if necessary) is often used to avoid unnecessarily exposing patients to side effects attributed to higher efficacy drugs (Le Page and Edan, 2018). We therefore set $r_t = c_t \lambda$, where $\lambda$ is the constant incremental risk associated with moving up the ladder of efficacy (which is set by the user). $c_t$ takes on a value of 0 for placebo, 1 for LE, 2 for ME, and 3 for HE. We define risk-adjusted CATE, as

$$\hat{\tau}_t^*(x) = \hat{\tau}_t(x) + r_t. \tag{4}$$

Assuming negative CATE indicates benefit, here a reduction in NE-T2 lesions, the tool then recommends treatment $j$ such that $j = \arg\min_t \hat{\tau}_t^*(x)$.

## 3. Experiments and Results

### 3.1. Dataset

The dataset is composed of patients from four randomized clinical trials: BRAVO (Vollmer et al., 2014), OPERA 1 (Hauser et al., 2017), OPERA 2 (Hauser et al., 2017), and DEFINE (Havrdova et al., 2013). Each trial enrolled patients with relapsing-remitting MS (the most common form) and had similar recruitment criteria. We excluded patients who did not complete all required MRI timepoints, or were missing MRI sequences/clinical features at baseline, resulting in a dataset with $n = 1817$. Treatments for these trials are categorized based on their efficacy at the group level: placebo ($n = 362$), no efficacy (NE, $n = 261$), lower efficacy (LE, $n = 295$), moderate efficacy (ME, $n = 431$), and high efficacy (HE, $n = 468$) with each level representing one treatment. Pre-trial statistics and treatment distributions can be seen in Appendix F.

All trials acquired MRIs at 1 x 1 x 3 mm resolution at the following timepoints: baseline (prior to treatment initiation), one year, and two years. Each contains 5 sequences: T1-weighted, T1-weighted with gadolinium contrast agent, T2-weighted, Fluid Attenuated Inverse Recovery, and Proton Density weighted. In addition, expert-annotated gadolinium-enhancing (Gad) lesion masks and T2 lesion labels are provided. The baseline MRIs and lesion masks were used as input to our model, while the NE-T2 lesion counts occurring between year one and two were used to compute count target and the binarized MEDA-T2 outcome. Patient's who did not complete all the required MRIs were excluded as they would not have a NE-T2 count. Percentage of MEDA-T2 in our dataset for placebo, NE, LE, ME, and HE are is 45.7%, 54.4%, 63.8%, 77.4%, 99.6%, respectively. In addition, baseline age, sex, and Expanded Disabillity Status Scale (Kurtzke, 1983), a clinical disability score, were

used as additional clinical features as inputs to our model. The dataset was divided into a 4x4 nested cross validation scheme for model evaluation (Krstajic et al., 2014). Following Soltys et al. (2014)'s use of ensembling, the 4 inner-fold models are used as members of an ensemble whose prediction on the outer fold's test set is the average of its members.

### 3.2. Predicting Future Lesion Suppression

We conduct three experiments to determine the best performing framework for predicting the observed future MEDA-T2 given different combinations of inputs, targets, and loss functions. The first compares the performance of the proposed single multi-head architecture with the performance of $m$ independently trained networks. The second assesses the benefit of using both imaging and clinical features. The third compares binary classification of MEDA-T2 with binarization of the output of a regression model trained directly on the NE-T2 lesion counts.

Table 1: Average precision scores for the binary MEDA-T2 outcome.

| Model Type | + Clinical | Multi-Head | Placebo AP | NE AP | LE AP | ME AP | HE AP |
|---|---|---|---|---|---|---|---|
| Random Baseline | | | 0.457 | 0.544 | 0.638 | 0.774 | **0.996** |
| Clinical Only | ✓ | | 0.72 +/- .08 | 0.76 +/- .02 | 0.82 +/- .06 | 0.90 +/- .03 | 0.995 +/- .01 |
| Binary Classification | ✓ | | 0.78 +/- .04 | 0.76 +/- .06 | 0.79+/-.03 | **0.916**+/-.02 | 0.997 +/- 0.01 |
| Binary Classification | | ✓ | 0.71 +/-.09 | 0.70 +/-.01 | 0.82 +/-.05 | 0.9 +/-.01 | 0.995 +/-.01 |
| Binary Classification | ✓ | ✓ | 0.78 +/-.08 | 0.79 +/-.03 | 0.86+/-.04 | 0.9 +/-.04 | 0.995 +/-.01 |
| Binarized Regression | ✓ | ✓ | **0.80** +/- .08 | **0.79**+/- .01 | **0.87** +/- .04 | 0.913+/- .03 | 0.996 +/-.01 |

Table 2: MSE for log lesion count regression against baseline (mean log lesion count).

| Model | Placebo | NE | LE | ME | HE |
|---|---|---|---|---|---|
| Baseline | 1.273 | 1.311 | 1.0432 | 0.904 | 0.0443 |
| Regression | 0.669 | 1.062 | 0.849 | 0.701 | 0.0433 |

Model performance is evaluated using average precision (AP) due to class imbalances in some of the treatment arms, particularly on HE. The random baseline reflects the positive MEDA-T2 label fraction on each arm. For an improved estimate of the generalization error, metrics are computed from the aggregated outer fold test set predictions. Results are shown in Table 1. The multi-head architecture improves APs across most treatment arms, and the concatenation of clinical features provides an additional boost in performance. Finally, the multi-head binarized regression model with clinical data concatenation outperformed the binary classification equivalent.

Given its strong performance, we performed the following evaluations using the regression model. We evaluated the MSE on the non-binarized output of the regression model (the log-lesion count), which demonstrates an improvement over the random baseline (mean log lesion count) for all treatments except HE (see Table 2). The failure to regress lesion counts on HE can be explained by the extremely small variance in the target distribution, with only 5% of all test patients having > 0 future NE-T2 lesion counts.

### 3.3. Estimating Treatment Effects

Given that the regression model outperforms alternatives on MEDA-T2 classification, and because it provides added granularity and a more intuitive interpretation, we used this model for CATE estimation. CATE estimates are computed for each treatment arm relative to placebo.

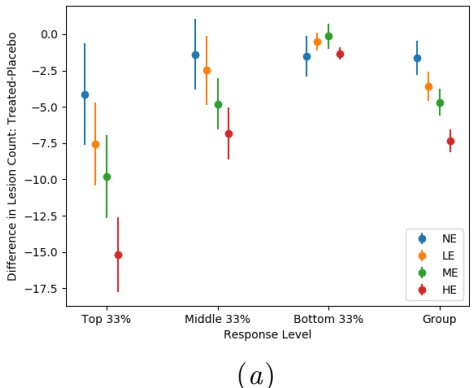
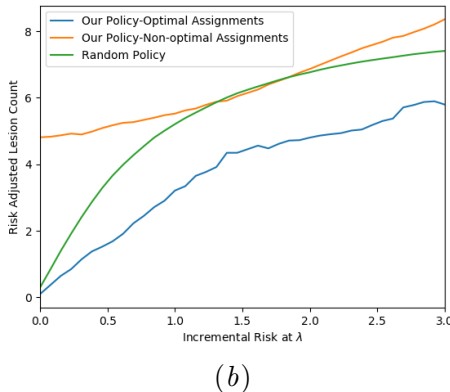

$(a)$ $(b)$

Figure 3: Treatment Effect Analysis. (a) Average lesion count differences between treatment-placebo pairs, binned according to tertiles of predicted treatment effect size. P-values for differences between groups are shown in Appendix G. (b) Average risk-adjusted lesion count for individuals who did (blue) or did not (orange) receive the recommended treatment, compared to random treatment assignment (green). Incremental risk values ($\lambda$) are varied on the $x$-axis.

To evaluate the quality of the CATE estimation, we report uplift bins (Ascarza, 2018) at three thresholds of predicted effect. Response ($\hat{\tau}_t$) values are binned into tertiles, and the average difference between the ground truth lesion count for patients who factually received the treatment $t$ and those who factually received placebo is computed for each treatment $t$. The result, shown in Figure 3$(a)$, demonstrates individuals predicted to respond most (top 33%) have a significantly greater reduction in lesion count over the entire group, and the ones predicted to respond least (bottom 33%) have a smaller reduction than the entire group. This suggests the model correctly identifies heterogeneous treatment effects. Uplift bins at different resolutions can be seen in Appendix D.

### 3.4. Clinical Decision Support Tool In Action

We now illustrate how the tool could be used in practice. Assuming each drug is associated with a different risk profile (see Section 2.3), Figure 4 illustrates examples of potential outcomes for two patients. Patient (a) might opt for either a HE efficacy option if they are not worried about greater risk of side effects or cost, or might select a ME option if they are more risk-averse. Patient (b), in turn, might opt for a drug that is NE at the group level but that is predicted to be of comparable efficacy to other options in their particular case.

Individual potential outcome predictions cannot be evaluated due to the lack of ground truth, but we can evaluate the group outcomes for those who received the recommended treatment. To do so, we adjust the ground-truth future NE-T2 lesion count for each individual who received the recommended treatment by adding the risk associated with that treatment, $y_i^* = y_i + r_t$, and compare their average risk-adjusted lesion count to the group who received a non-recommended treatment (Figure 3(b)). Patients who were factually assigned treatment based on the system's recommendation had a reduction in expected adjusted lesion count for any value of the incremental cost $\lambda$ (varied along the $x$-axis) which indicates the tool provides better treatment recommendations when minimizing treatment-associated risk.

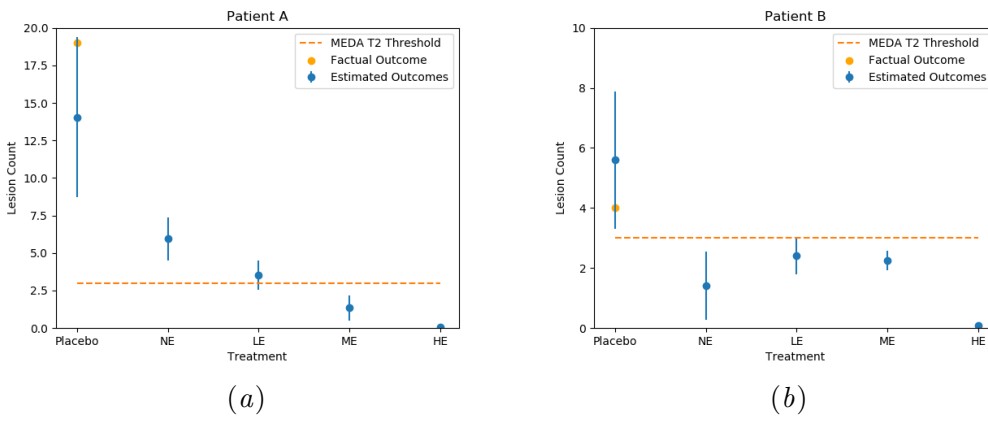

Figure 4: Predicted future lesion count on each treatment for two different test patients. Error bars indicate the standard deviation of the ensemble prediction. The MEDA-T2 threshold (3 lesions) is depicted by the dashed line.

## 4. Conclusions

In this paper, we introduce the first medical imaging-based deep learning model for recommending optimal treatments in MS. The model predicts future NE-T2 counts and MEDA-T2 with high precision on 5 different treatments, and finds sub-groups with heterogeneous treatment effects. However, highly effective suppression of new lesion formation may have only a modest effect on long term disability progression. Current work is focused on predicting stronger markers of disability progression, so as to improve the value of the decision support tool. Additionally, the model's recommendations have the potential to balance efficacy against treatment associated risks and patient preference. However, our current support tool uses linear scaling of risk between treatments. A comprehensive risk adjustment model that incorporates patient preferences, side effects, cost and other inconveniences would provide a more holistic clinical support tool but is beyond the scope of this paper. Future improvements could also be made by estimating treatment effect uncertainty (Jesson et al., 2020) and explicitly optimizing adjusted CATE (Zhao and Harinen, 2020).

## Acknowledgments

This investigation was supported (in part) by an award from the International Progressive Multiple Sclerosis Alliance (award reference number PA-1412-02420), the Canada Institute for Advanced Research (CIFAR) Artificial Intelligence Chairs program (Arbel), the Natural Sciences and Engineering Research Council of Canada (Arbel), an end MS Personnel Award from the Multiple Sclerosis Society of Canada (Falet), a Canada Graduate Scholarship-Masters Award from the Canadian Institutes of Health Research (Falet), and the Fonds de recherche Santé / Ministère de la Santé et des Services sociaux training program for specialty medicine residents with an interest in pursuing a research career, Phase 1 (Falet). Supplementary computational resources and technical support were provided by Calcul Québec, WestGrid, and Compute Canada. Additionally, the authors would like to thank Louis Collins and Mahsa Dadar for preprocessing the MRI data, Zografos Caramanos, Alfredo Morales Pinzon, Charles Guttmann and István Mórocz for collating the clinical data, Sridar Narayanan. Maria-Pia Sormani for their MS expertise, and Behrooz Mahasseni for many helpful discussions during model development.

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

## Appendix A. Implementation Details

The MRI sequences are first clipped between $+/-3$ standard deviations and then normalized to $N(0, 1)$ per sequence. The MRI sequences are then resampled to 2x2x2 resolution and cropped for a final dimension of 72x76x52. The clinical data is normalized to $N(0, 1)$.

As mentioned in the Network Architecture section, the trunk of the model consists of three ResNet blocks followed by several MLPs. Each ResNet block contains two convolutional blocks followed by a residual addition. Each convolutional block contains a convolution (kernel size 3, stride 1), Instance Normalization (Ulyanov et al., 2017), a dropout layer (Srivastava et al., 2014) with $p = 0.3$, and a LeakyReLU activation (Maas, 2013). Each ResNet block, with the exception of the last, is followed by an max pooling operation with kernel size 2. In the three ResNet blocks, the number of kernels for each convolution is [32, 64, 128] respectively. After the three ResNet blocks, the latents are flattened using a global average pool before concatenating the features with the clinical information and inputting the combined latent space to the MLPs. Each of the 5 MLPs in the network consist of three hidden layers which have dimensions [128,32,16] and use ReLU activations (Agarap, 2018) with no dropout. For training, we used the AdamW optimizer(Loshchilov and Hutter, 2019) with a learning rate of .0001 and a batch size of 8.

For models using imaging data and clinical data, the clinical data included age, gender and baseline EDSS. For the models using clinical data only, the clinical data included age, gender, baseline EDSS, baseline T2 lesion volume, and baseline Gad lesion count.

## Appendix B. Lesion Counts

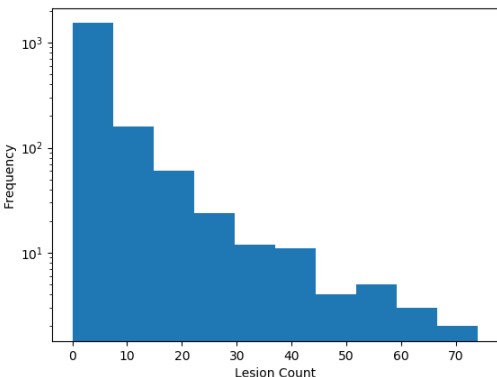

Figure B.1: Future NE-T2 Lesion Count Histogram.

Figure B.2: Future NE-T2 Lesion Counts by Treatment

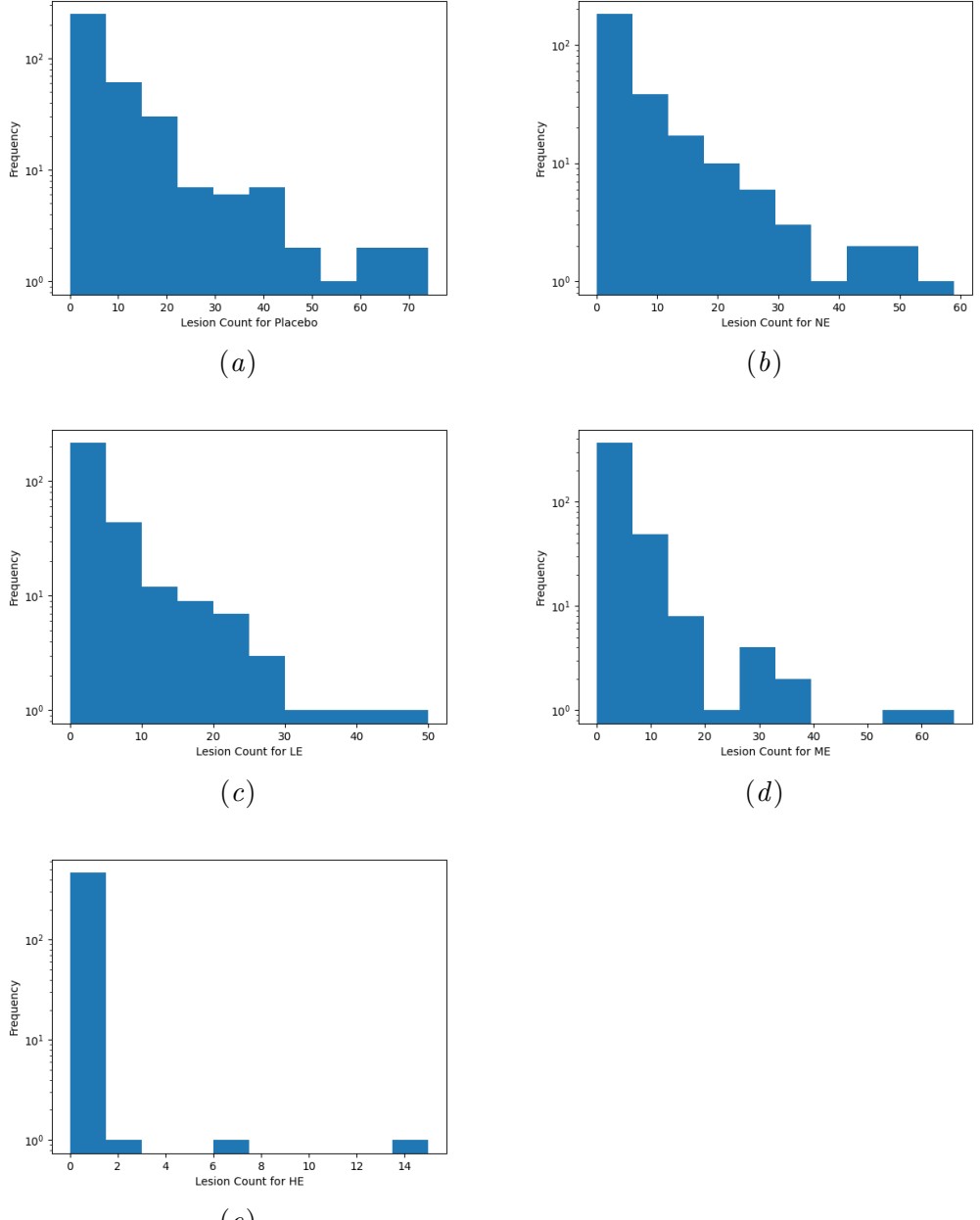

## Appendix C. Treatment Effect Analysis with the binary MEDA-T2 outcome

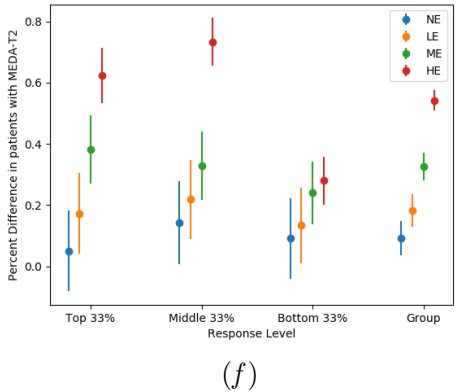
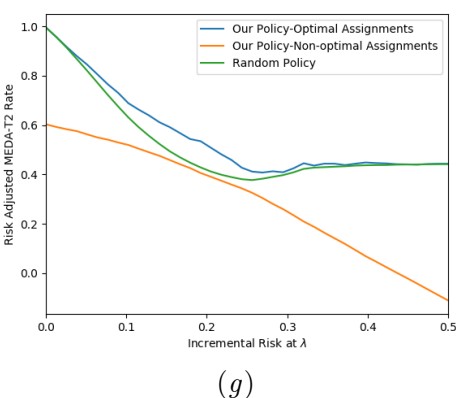

$(f)$          $(g)$

Figure C.1: Treatment Effect Analysis. (a) Average difference in frequency of MEDA-T2 between treatment-placebo pairs, binned according to tertiles of predicted treatment effect size. (b) Frequency of risk-adjusted MEDA-T2 for individuals who did (blue) or did not (orange) receive the recommended treatment, compared to random treatment assignment (green). Incremental risk values ($\lambda$) are varied on the $x$-axis.

## Appendix D. Additional Uplift Bins

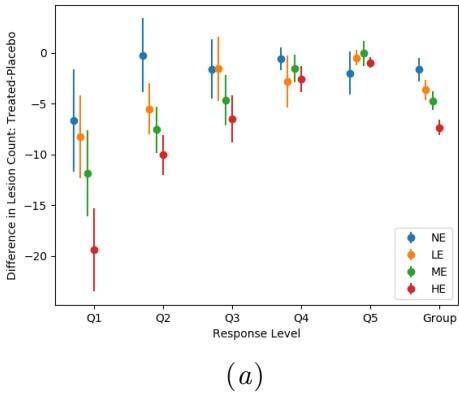
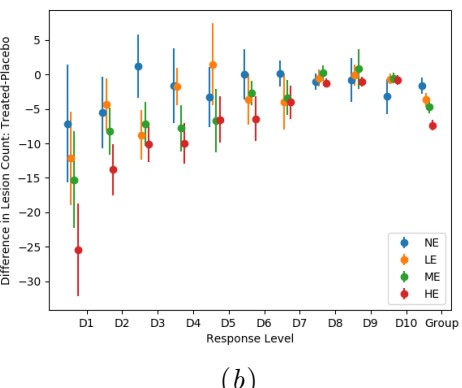

$(a)$          $(b)$

Figure D.1: Average difference in NE-T2 lesion count between treatment-placebo pairs, binned according to quintiles (a) and deciles (b) of predicted treatment effect size.

## Appendix E. Additional Results

Table E.1: ROC-AUC for the binary MEDA-T2 outcome.

| Model Type | + Clinical | Multi-Head | Placebo | NE | LE | ME | HE |
|---|---|---|---|---|---|---|---|
| Baseline | | | 0.5 | 0.5 | 0.5 | 0.5 | 0.5 |
| Clinical Only | ✓ | | 0.76 + .03 | 0.719+/-.02 | 0.745+/-.05 | 0.73+/-.03 | 0.46+/- .07 |
| Binary Classification | ✓ | | 0.77 + .03 | 0.69+/-.05 | 0.68+/-.06 | 0.759+/-.03 | 0.5 +/- .11 |
| Binary Classification | ✓ | ✓ | 0.818 +/- .01 | 0.738 +/- .071 | 0.770 +/- .001 | 0.753 +/- .014 | 0.488 +/- .0017 |
| Binary Classification | | ✓ | 0.772 +/- .04 | 0.682 +/- .04 | 0.73 +/- .01 | 0.751+/- .04 | 0.497+/- .04 |
| Binarized Regression | ✓ | ✓ | **0.836 +/- .01** | **0.749 +/- .0021** | **0.783 +/- .001** | **0.769 +/- .014** | 0.488 +/- .0017 |

Table E.2: MAE for log lesion count regression against baseline

| Model | Placebo | NE | LE | ME | HE |
|---|---|---|---|---|---|
| Baseline | 0.94 | 0.98 | 0.89 | 0.789 | 0.072 |
| MAE | 0.658 +/- .08 | 0.839 +/- .059 | 0.70 +/-.052 | 0.64 +/- .07 | 0.07+/- .01 |

## Appendix F. Pretrial Patient Statistics.

Table F.1: Baseline clinical and scalar MRI metrics for our dataset. Standard deviations are in parentheses.

| Trial/Treatment | BRAVO/Placebo | DEFINE/Placebo | BRAVO/NE | BRAVO/LE | OPERA 1/ME | OPERA 2/ME | OPERA 1/HE | OPERA 2/HE |
|---|---|---|---|---|---|---|---|---|
| N | 278 | 94 | 261 | 295 | 223 | 208 | 236 | 232 |
| Age | 37.95 (9.27) | 37.8 (9.51) | 37.03 (9.2) | 38.29 (9.45) | 37.2 (9.25) | 37.3 (8.95) | 37.1 (9.27) | 37.5 (8.85) |
| Gender(Male Fraction) | 0.29 | 0.25 | 0.29 | 0.31 | 0.33 | 0.34 | 0.34 | 0.37 |
| Baseline EDSS | 2.71 (1.16) | 2.46 (1.23) | 2.67 (1.23) | 2.64 (1.14) | 2.7 (1.27) | 2.68 (1.37) | 2.77 (1.21) | 2.68 (1.27) |
| T2 Lesion Volume | 7.82 (8.714) | 6.67 (8.2) | 9.28 (9.8) | 8.4 (9.2) | 9.28 (11.1) | 10.0 (12.3) | 10.96 (14.21) | 10.83 (14.25) |
| Gad Count | 1.12 (3.24) | 1.84 (3.91) | 1.61 (4.40) | 1.48 (3.5) | 1.535 (4.75) | 1.87 (4.47) | 1.73 (4.35) | 1.85 (4.8) |

Table F.2: Treatments used for the model by trial.

| Trial | High Efficacy Treatment | Moderate Efficacy Treatment | Lower Efficacy Treatment | No Efficacy Treatment | Placebo |
|---|---|---|---|---|---|
| OPERA 1 | Ocrelizumab | INFb-1a SC | | | |
| OPERA 2 | Ocrelizumab | INFb-1a SC | | | |
| BRAVO | | | Avonex | Laquinimod | Placebo |
| DEFINE | | | | | Placebo |

## Appendix G. Significance Values

Table G.1: P values for group differences between response groups for the regression task shown in figure Figure 3(a). Column headers indicate the two responder groups (tertiles) that are being compared.

| Grouping | NE | LE | ME | HE |
|---|---|---|---|---|
| Top 33%-Middle 33% | .0043 | <.001 | <.001 | <.001 |
| Top 33%-Bottom 33% | .0028 | <.001 | <.001 | <.001 |
| Top 33% -Group 33% | .031 | <.001 | <.001 | <.001 |
| Middle 33%-Bottom 33% | .89 | .03 | <.001 | <.001 |
| Middle 33%-Group 33% | .70 | .19 | .97 | .43 |
| Bottom 33%-Group 33% | .78 | <.001 | <.001 | <.001 |

## Appendix H. MRI Preprocessing

Scans were first denoised (Manjón et al., 2010), corrected for intensity heterogeneity (Sled et al., 2002), and normalized into the range 0-100. Second, for each patient, the T2w, PD, and FLAIR scans were co-registered to the structural T1w scan using a 6-parameter rigid registration and a mutual information objective function (Collins et al., 1994). The T1w scans were then registered to an average template defining stereotaxic space (Collins and C. Evans, 2011; Fonov et al., 2011). All volumes are resampled onto a 1 mm isotropic grid using the T1-to-stx space transformation (for the T1w data) or the transformation that results from concatenating the contrast-to-T1 and T1-to-stx transformation (for the other contrasts).

