# OpenReview forum: "Personalized Prediction of Future Lesion Activity and Treatment Effect in Multiple Sclerosis from Baseline MRI"
_MIDL.io/2022/Conference — MIDL 2022_

### Official Review · Reviewer_4LeE · 2022-01-24

**Confidence:** 5
**Preliminary Rating:** 5
**Recommendation:** Best Paper Award, Oral

**Summary:**

This paper presents a multi-head deep neural network model for individualized treatment decisions in MS, to predict futures new and enlarged lesion counts, and the conditional average treatment effect (CATE), as defined by the predicted future suppression of NE-T2 lesions, between different treatment options relative to placebo. Dataset is a large, proprietary, federated dataset of 1817 MS patients. Used method is a ResNet encoder with multiple decoders using causal inference as objective in clinical trial data for different treatments. Two tasks are evaluated: prediction lesion count (regression) and treatment effect (classification). Results give convincing insight in what treatment strategy is optimal for an individual patients.

**Strengths:**

- Paper makes me enthusiastic! I appreciate the clinical relevance, novel approach, writing style and validation experiments.
- Very unique dataset of randomized controlled trials in MS
- Ambitious goal: prediction lesion counts and treatment effect
- Extensive and clear review of related literature to provide context.
- First image-based treatment recommendation framework for MS that combines the notion of predicting prognosis, treatment effect estimation, and treatment-associated risk
- Another strong and novel point is the derivation of the deep learning objective from causal estimation of treatment effect in clinical trials -> CATE-estimation.
- The patient specific risk profiles are a strong output which may be very clinically useful.
- The additional analyses of CATE and individual profiles is very strong.

**Weaknesses:**

I have only minor weaknesses to mention here:
- Differences in MR protocol between clinical trials is not addressed.
- The precision metric is unclear/may provide an incomplete picture of model performance
- Patient outcome is based on MRI only, it could be stronger if outcomes related to disability and quality of life could be predicted as well.


**Deanonymize Review:**

yes

**Detailed Comments:**

- I had to look back a couple of time to remind myself of the MEDA, CATE, CDST abbreviation meaning. Try to reduce the number of new abbrevations to improve readability.

**Paper Type:**

methodological development

**Questions To Address In The Rebuttal:**

- What were differences between clinical trials in MRI acquisition? Can studies be merges without harmonization?
- Is specific clinical trial a confounder in this analysis? Which type of drugs (NE, LE, ME, HE) was evaluated in which trial?
- Prediction performance was quantified using precision as a metric, which may not give a complete picture of performance. AUC, balanced accuracy, or the combination of precision and recall would be better. Please provide recall or AUC in addition to precision.
- Also only the average precision is given in Table 1, please provide stdev (or confidence interval) as well.


**Special Issue:**

yes

---

### Official Review · Reviewer_gkt4 · 2022-01-25

**Confidence:** 4
**Preliminary Rating:** 5

**Summary:**

Authors proposed a method for predicting the MS treatment outcomes in terms of new or enlarged T2 lesions counts from baseline brain MRIs. They further developed the prediction models into an image-based treatment recommendation framework for MS. The proposed approached was trained and tested using real-life RCT data.

**Strengths:**

- use of multi-sequence MRI
- use of a multi-head / multi-taks DL architecture
- inclusion of multiple MS treatments as outcomes
- data composed of multi-trial, multi-scanner dataset of MS patients
- real clinical trial data used in modeling and testing
- incorporation of clinical data into the DL architecture
- comparison of both regression and categorical models


**Weaknesses:**

- One of the conclusions is "this framework shows that improved lesion suppression can be achieved using an appropriate CDST, especially when treatment risk is being considered". This will require a prospective study to test the proposed CDST. I think this is out of scope of this paper.


**Deanonymize Review:**

no

**Paper Type:**

validation/application paper

**Questions To Address In The Rebuttal:**


- why were participants missing MRI timepoints excluded? only pre-treatment
- did RCT all have the same treatment - e.g. why to assume treatment response being independent of the mechanism of the treatment? what is the justification for pooling data from different RCTs and collapsing the outcomes as low/moderate/high efficacy?
- should placebo considered a separate arm or similar as no efficacy?


**Special Issue:**

yes

---

### Official Review · Reviewer_3Xzd · 2022-01-26

**Confidence:** 4
**Preliminary Rating:** 4
**Recommendation:** Oral, Poster

**Summary:**

In this work, the authors propose a deep learning framework to predict treatment effect in Multiple Sclerosis in terms of lesion activity while accounting for treatment-associated risk. The proposed method is validated on a large (n=1817) multi-centric MS cohort. They show that MRI, together with clinical data, can predict the treatment efficacy for five treatment types.

**Strengths:**

This paper addresses the important topic of treatment planning, which is the fundamental basis of precision medicine. In the last decades, extensive research has focused on the extraction and definition of novel radiological biomarkers to allow a better characterization of the disease. However, predictive models to support treatment planning decision, like the one described in this work, have been poorly explored so far.

Another strength of this work is the dataset used, which not only is very large (n=1817), but includes multi-centric and longitudinal data.


**Weaknesses:**

Due to the complexity of the task addressed in this work, this work includes some oversimplifications that might lead to misleading interpretation. In particular, lesion activity is the only hallmark used to evaluate treatment efficiency. However, MS is known to be also characterized by other pathophysiological mechanisms such as more diffuse inflammation and atrophy. Moreover, due to a clinico-radiological paradox, it is well known that patients with large lesion loads are not always associated with worse clinical outcomes (Barkhof 2002). Therefore, the clinical outcome of a treatment is also to be taken into account for the treatment decision, in addition to radiological evaluations.
Another simplification that is made, is to group drugs into three groups (low/moderate/high efficiency), with an incremental risk factor between the categories. It is up to the user to define this incremental constant, but the impact of this choice is not discussed in the presented work.

Further, the impact of clinical data (demographics, clinical scores...) on the predictions is poorly explored. Whilst the authors compare their model to an equivalent without clinical data, the comparison with a model solely based on clinical data is not shown, and could further confirm the relevance of this work. Also, the demographics distribution between the treatment groups (placebo, NE, LE, ME, HE) is not shown. To ensure that no systematic differences in between groups are biasing the prediction model, age and EDSS matched subgroups of patients should be used. Similarly, the distribution of NE-T2 lesions in the different groups is not shown, and could inform on the effect size of the predictions.

Finally, the authors should better identify and discuss the limitations of their work

**Deanonymize Review:**

no

**Detailed Comments:**

Some minor points to be addressed

- In section 2.1, "the remaining are m-1 treatment options", should be m instead

- In section 2.2, "Specifically, we a single..", missing verb

- In figure 2, the authors could also report the mean average error, to provide a more intuitive sense of the scale of the estimated error

- When reporting the results shown in Figure 3, the authors claim that the values are different between the response levels, however no statistical test is performed.

- When reporting the average precision of the different models in Table 1 and 2, the authors could consider to perform a repeated cross-validation to be able to assess the significance level of the AP difference between models.

- In figure 4, it is not clear to me what the MEDA T2 threshold is

- Please report and formally compare the demographics distribution and NE-T2 lesion counts for the different treatment groups

- The data used included two followup time points, after 1 and 2 years respectively. How were these two timepoints used for each patient ? How did the authors account for the time elapsed between two scans ?

**Final Rating After The Rebuttal:**

5: Strong Accept

**Justification Of The Final Rating:**

The authors deepened their analysis on the impact of clinical variables on treatment predictions, which was my major critique on this work.
They now report the distribution of clinical variables within the different patient groups and compare their predictive models to a baseline using only clinical information.

**Paper Type:**

both

**Questions To Address In The Rebuttal:**

The major points to be addressed in the rebuttal are:

- the distribution of demographics in the different treatment groups and the possible impact of such differences in the prediction

- how is time considered when using multiple follow-up scans ? Are you also considering the occurrence of NE-T2 lesions after 2 years (i.e.,  year 2 versus baseline) ?

- discuss the limitations of this work, in particular concerning the choice of the marker of disease evolution (i.e., lesion activity), treatment classification and risk increment implementation

**Special Issue:**

no

---

### Meta-Review · Area_Chair_NVp7 · 2022-02-19

**Recommendation:** Accept (Oral)
**Confidence:** 5

**Metareview:**

The paper proposes to combine MR image embeddings from a ResNet-style network with clinical data, with the prognostic aim to predict future lesion counts and disease activity. A set of treatment-specific MLPs are trained on the joint (image, clinical data) representation to this end. The approach is evaluated in a multi-centric setting on four large clinical trials.

The work is well done, and has clear real-world impact for future clinical trials. The treatment-specific aspect is a good way to deal with what is otherwise a major confounder in such studies, although one can imagine that it would be difficult to train with smaller datasets.

The revised version has improved readability, e.g. acronym usage is reduced.

Pros
* large, multicentric dataset
* treatment-specific modeling improved real-world relevance

Cons
* proprietary dataset
* focus on lesion count is traditional and expected by clinicians, but may hide more subtle and relevant disease effects, in particular in normal-appearing white matter

---

### Decision · Program_Chairs · 2022-02-28

Accept